# Examination of factors associated with the temporal stability assessment of crash severity by using generalised linear model—A case study

**Abdulaziz H. Alshehri[1], Amjad Pervez[2], Muhammad Hussain[3]\*, Danish Farooq[4], Etikaf Hussain[5]**

**1** Department of Civil Engineering, College of Engineering, Najran University, Najran, Saudi Arabia, **2** School of Traffic and Transportation Engineering, Central South University, Changsha, China, **3** Western Australian Centre for Road Safety Research, School of Psychological Science M304, Perth, Western Australia, Australia, **4** Civil Engineering Department, COMSATS University Islamabad, Wah Campus, Islamabad, Pakistan, **5** Veltch Lister Consulting, Level 5, Brisbane, Queensland, Australia

\* muhammad.hussain@uwa.edu.au

**Data Availability Statement:** All relevant data are within the paper.

**Funding:** The authors are thankful to the Deanship of Scientific Research at Najran University for funding this work under the General Research

## Abstract

Road crashes are a major public safety concern in Pakistan. Prior studies in Pakistan investigated the impact of different factors on road crashes but did not consider the temporal stability of crash data. This means that the recommendations based on these studies are not fully effective, as the impact of certain factors may change over time. To address this gap in the literature, this study aims to identify the factors contributing to crash severity in road crashes and examine how their impact varies over time. In this comprehensive study, we utilized Generalised Linear Model (GLM) on the crash data between the years 2013 to 2017, encompassing a total sample of 802 road crashes occurred on the N-5 road section in Pakistan, a 429-kilometer stretch connecting two big cities of Pakistan, i.e., Peshawar and Lahore. The purpose of the GLM was to quantify the temporal stability of the factors contributing crash severity in each year from 2013 to 2017. Within this dataset, 60% ($n = 471$) were fatal crashes, while the remaining 40% ($n = 321$) were non-fatal. The results revealed that the factors including the day of the week, the location of the crashes, weather conditions, causes of the crashes, and the types of vehicles involved, exhibited the temporal instability over time. In summary, our study provides in-depth insights aimed at reducing crash severity and potentially aiding in the development of effective crash mitigation policies in Pakistan and other nations having similar road safety problems. This research holds great promise in exploring the dynamic safety implications of emerging transportation technologies, particularly in the context of the widespread adoption of connected and autonomous vehicles.

Funding program grant code (NU/DRP/SERC/12/55). The first author (Abdulaziz H. Alshehri) is the primary recipient of this funding.

**Competing interests:** The authors have declared that no competing interests exist.

## Introduction

Road safety is a continually growing concern within any contemporary society due to ever-rising road crashes resulting in significant number of fatalities and injuries. According to the recent report of World Health Organisation (WHO), there were an estimated 1.19 million road traffic deaths in 2021 world widely. Road crashes are the 1st leading cause of the deaths of children and young aged 5–29 years, and 12th leading cause of deaths of all ages [1]. Road crashes have an adverse impact on the health and well-being of injury survivors and their families. Therefore, it is vital to monitor road safety progress regularly to improve road safety and sustainability within the transportation system [2,3]. Road crashes are complex and multi-faceted events that occur due to the interactions of various factors [4], such as driving behaviour [5–11], road and vehicle conditions [12], weather conditions [13,14], road geometry and spatial features of the road environment [15] and many others. By developing effective road safety policies and interventions, the likelihood and severity of crashes could be significantly reduced.

Most statistical analyses of highway safety data assume that the factors affecting crashes remain stable over a period of time. This assumption may not reflect the dynamic nature of road safety issues, and therefore, it is important to consider the temporal stability of the contributing factors in road safety research. The temporal stability could also be crucial in illuminating crash trends, and could have significant repercussions for conventional statistical analyses that use data to estimate parameters for different explanatory variables to ascertain the impact of these variables on the likelihood and resulting crash severities (parameters that are typically assumed to be fixed over time) [16]. By doing so, policymakers can ensure that they are using up-to-date information to allocate resources effectively and improve road safety outcomes.

Nine out of ten deaths occur in developing (low and middle-income) countries due to road crashes, while people in low-income countries continue to face the highest risk of fatalities per 100,000 population [1]. Developed countries adopt the 3Es policy tool (Education, Engineering, and Enforcement), to strengthen the road transport system's three components: roads, drivers, and vehicles, and improve road safety. But developing countries are lacking this approach to fully implement, therefore, they are suffering from road trauma more than developed countries. It's crucial to recognise that the effects of various factors on crash likelihood and severity might exhibit fluctuations over time [17]. Recent studies have attempted to provide the temporal variations in road crashes or crash severities over a dispersed period in various developed and developing countries [18–22]. These studies investigated the temporal instability of factors by estimating yearly effect instead of estimating an average effect in the total period. Analysing temporal stability helps identify if crash rates or severity follow consistent trends over time or if they vary, which can be crucial for implementing effective road safety measures and policies.

Crash data is often analysed to determine the factors impacting different crash outcomes, particularly crash severity. This analysis categorises crash severities into discrete outcomes, using models like ordered (probit and logit models) and unordered models (such as multinomial and mixed logit models), which are frequently employed in crash severity analysis [23]. For instance, different modelling techniques have been employed by different researchers such as random parameter hierarchical ordered logit model [24], ordered probit model [25], random-effects generalised ordered probit model [26], random parameter logit approach [27], and random parameter ordered logit model [28]. Among all techniques, ordered probit and logit models are preferred because of the nature of data (ordinal). Washington et al (2020) argued that the traditional form of these models can restrict what variables affect probabilities

of outcomes, leading to incorrect conclusions [29]. Therefore, the most recent crash severity studies used mixed logit modelling approach.

Pakistan, being lower-middle income country, is facing serious road safety challenges, with approximately 26,811 people dying in 50,283 recorded crashes from 2016 to 2020 according to Pakistan Bureau of Statistics data [30]. The overall ratio of fatalities in road crashes has reached 55%, the highest in the country's history. Pakistan's fatality rate is 14.2 per 100,000 vehicles, significantly higher than the UK's rate of 3.2 per 100,000 vehicles, despite the UK having six times more registered vehicles [31]. The issue of road crashes is threatening public health and GDP. Despite implementing fundamental road safety measures such as appropriate road design, traffic management, speed regulations, and the promotion of seat belt and helmet use, the inadequacy of law enforcement and modern traffic monitoring systems—like high-definition cameras, advanced driving safety technologies, fatigue/distraction detection systems, and night vision systems—aggravates road safety concerns. In 2016, the Government of Pakistan launched the "Road Safety Pakistan" project, which developed the National Road Safety Strategy 2018–2030, based on the safe system approach devised by the WHO. The strategy sets out a long-term vision to improve safety on national, provincial, and local roads, with the aim of saving more than 6000 lives by 2030. Fig 1 outlines the strategy devised by the Government of Pakistan, inspired by the safe system approach from WHO, considering the causes of road crashes and the developed policy tools.

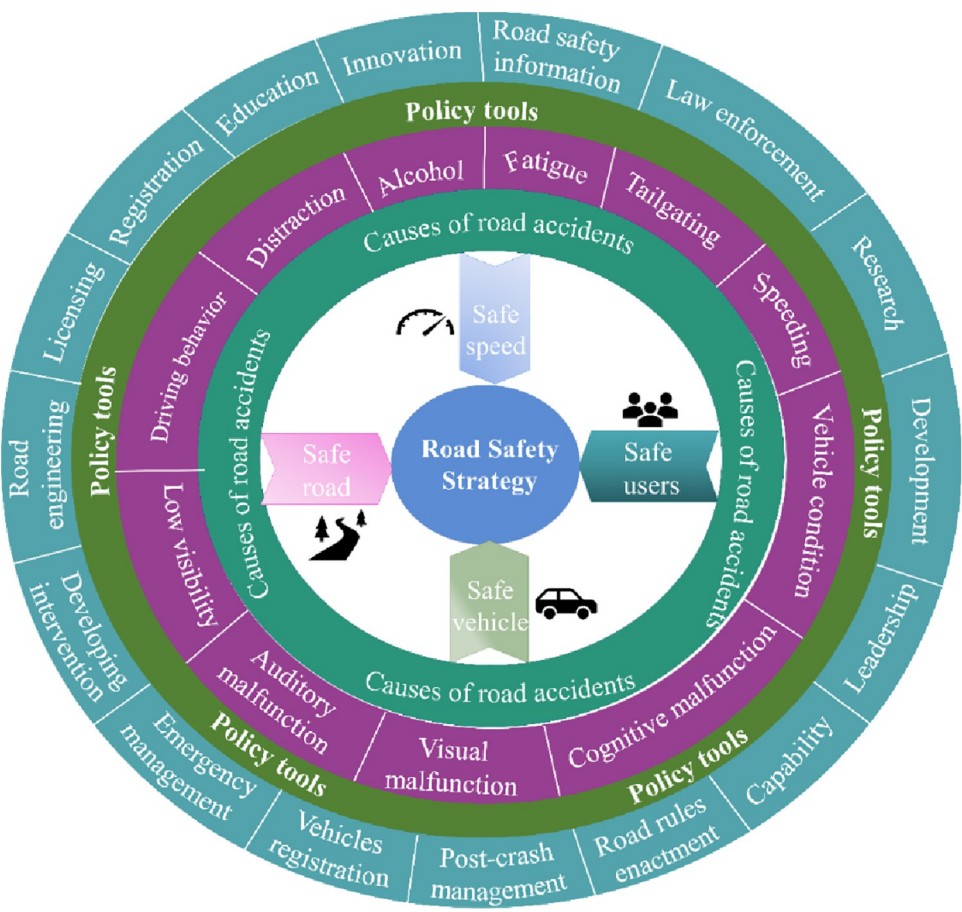

**Fig 1. Road safety approach implemented in Pakistan.**

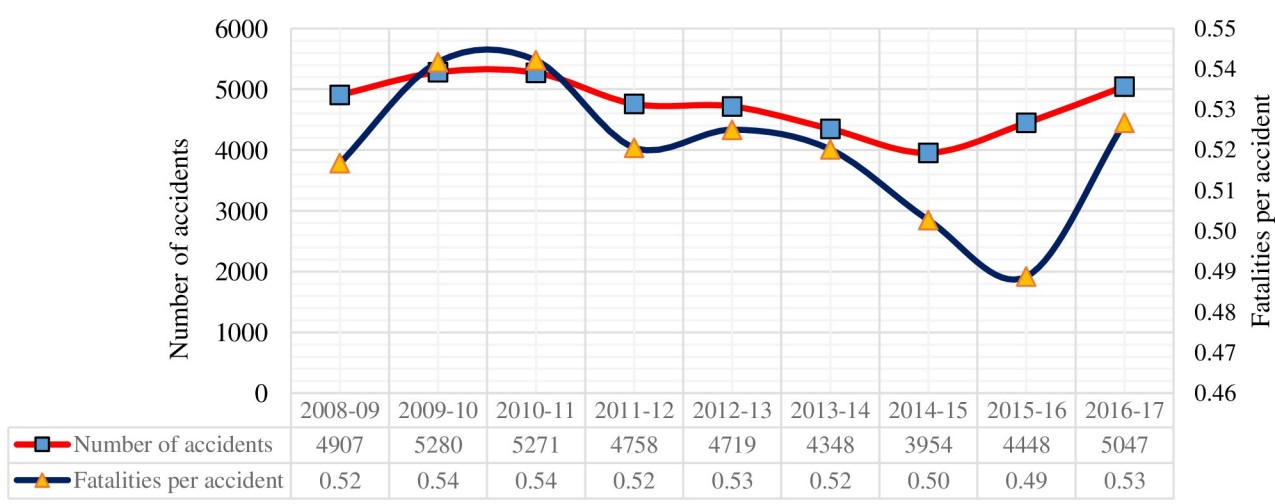

**Fig 2. Comparison between the number of crashes and fatalities per crash on the road system in Pakistan (2008–2017).**

Interestingly, road crashes in Pakistan are experiencing a slight decreasing trend in last couple of years. This reduction might be attributed to several policies and measures implemented by the government as mentioned in Fig 2. Effective road safety measures such as the up gradation of roads, speed limit enforcement, and promoting seatbelts have been implemented in Pakistan, resulting in a reduction in the number of road crashes. However, despite this progress, the fatalities per crash continue to show a non-linear trend in last couple of years (see Fig 2). To tackle this issue effectively, the emphasis should shift from merely reducing crash frequency to comprehending crash severity and its temporal consistency over time. This shift is vital for developing targeted countermeasures aimed at enhancing road safety in Pakistan.

In our previously published articles, factors contributing to road crashes among Pakistani drivers were thoroughly investigated [5–7]. However, to the best of our knowledge, the factors associated with the temporal stability of crash severity in Pakistan has rarely been investigated in the literature. Building on this literature gap, the current study employed GLM on the five years road crash data (2013–2017), collected from National Highway and Motorway Police (NHMP) and examined the factors contributing to crash severity. The data used in this study is rich enough to provide valuable findings to help create road safety improvement policies. The findings of this study are beneficial for policymakers and government agencies to administrate the traffic safety laws and provide a safe transportation system in the country.

## Research design

### Study area

In Pakistan, one of the roads with the highest fatality rate is the national highway N-5, with rural, urban, and interurban traffic. The National Highway 5 (N-5) is a 1819 km national highway in Pakistan, which extends from two big cities of Pakistan; Karachi and Peshawar. It is the longest national highway in Pakistan and serves as an important North-South Road artery [32]. Being the largest and the busiest national highway of Pakistan, N-5 renders its commuters at high risk of getting involved in crashes. For this study, the N-5 road section between Peshawar and Lahore (429 km), Pakistan, has been selected, with objectives to explore the trends and patterns, of road crashes. The geo-referenced coordinates for N-5 from Lahore to Peshawar are as follows: from (latitude 32.831867˚, longitude 73.833233˚) to (latitude 33.086486˚,

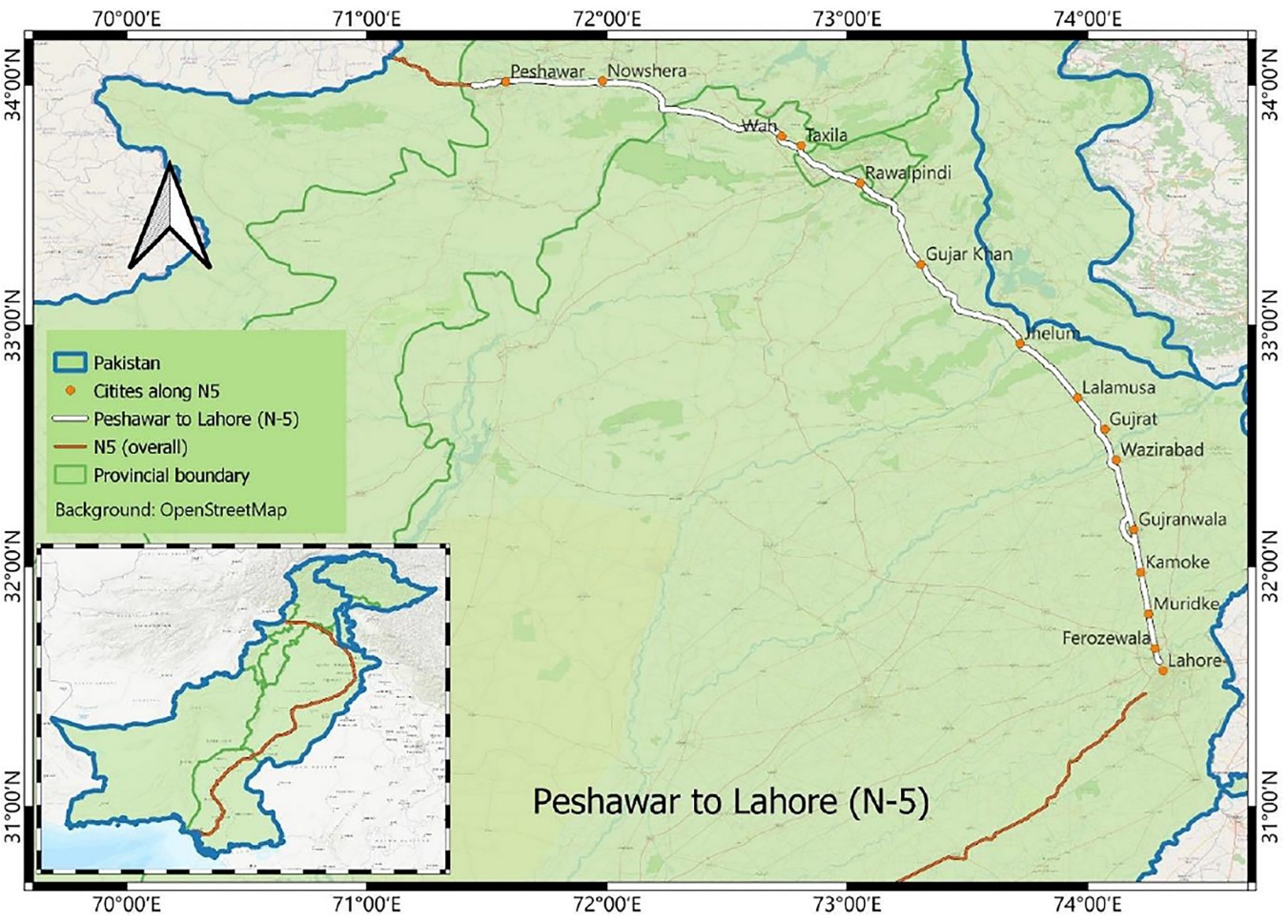

**Fig 3. N-5 alignment between Peshawar and Lahore, Pakistan.**

longitude 73.447178˚). Fig 3 illustrates the alignment of the N-5 road section in Pakistan, delineating the study area.

### Data collection

This study utilises historical crash data of N-5, collected from the NHMP, Pakistan, which is the only legitimate source of road crash data in Pakistan. The road crash data obtained represents five years of road crashes, from 2013 to 2017. More recent data was not available in a consistent format at the time of this research. The second reason for the inclusion of road crashes data between 2013–2017 is that there is no major infrastructure developed on N-5 after the mentioned years, so it is believed that a same trend of road crashes on N-5 is followed [32].

The dataset includes the number of crashes, number of fatalities and injuries in each crash, day, time, and weather of the occurrence of crashes, beat number (the road section where the crashes occurred), cause of specific road crashes, and nature of vehicles involved in road crashes. Days has been classified from Monday to Sunday. Time has been classified into two groups according to peoples' working span and lifestyle in Pakistan such as: day timing (0600–01800), and night timing (1800–0600). According to the Bureau of Meteorology Department, Pakistan, the country has four different seasons, such as spring (March-May), summer (June–

August), autumn (September–November), and winter (December–February). Therefore, we selected spring and autumn season as mild, summer as hot and winter as cold. Beat number has been assigned from Beat 1 to Beat 11. Beat number refers to the specific patch of N-5 covering the whole study area (N-5 from Peshawar to Lahore) by numbering starts from Beat 1 at Peshawar and end at Beat number 11 in Lahore. By reviewing the literature, we considered all the factors contributing to road crashes. However, we shortlisted several contributory factors by considering the local context (see S1 Table). The nature of vehicles has been classified with in three different groups: Light-Traffic Vehicles (LTVs) such as cars, pickups, taxies etc), Heavy-Traffic Vehicles (HTVs) such as buses, trucks, trailers), and Three Wheeled vehicles (3Ws) such as bicycles, rickshaws, and motorcycles.

To assess the actual cost of a crash and to investigate the impact of the crash on human life, crash severity plays a crucial role in the crash analysis. Severity analysis evaluates the safety efficiency of the road transportation system. Crash severity is a dimensionless value that can be defined as *"the ratio of the number of fatalities to the number of road crashes in unit time"* [33].

## Methodological approach

As mentioned in first section of the paper, there are many different methodological approaches being used in the literature. However, in this study, the GLM is used for the temporal stability of factors affecting the crash severity of road crashes in Pakistan. The reason is associated with the fact that dependent variable is continuous and the GLM allows to investigate the effects of independent variables on dependent variables having continuous nature. In GLM, the dependent variables and the independent variables are assumed to have variance following a single-parameter exponential family of probability distribution [34]. The standard linear regression has multiple underlying assumptions, including (i) all observed cases in the dependent variables follows a normal distribution, i.e., $y_i \sim N(\mu_i, \sigma_i^2)$; (ii) In all observations, the variance of distribution is same, i.e., $\sigma_i^2 = \sigma^2$; (iii) A direct relation exists between the dependent variable and the expected model value, i.e., $E(x_i\beta) = \mu_i$. Here, $\mu$ is the mean value, $x$ is the independent variable, and $\beta$'s is the respective co-efficient. The primary advantage of GLM is that it relaxes the assumptions of linear model by restructuring the predictor and the fit [35].

In addition, GLM are characterised by following attributes: (i) Distribution of variance of the random component of independent variable, $y$, belongs to exponential family; (ii) A linear relation between the linear predictor, $\eta$, and the product of design matrix $X$ and $\beta$ such that, $\eta = X\beta$; (iii) A known one-to-one differentiable link function $g(\cdot)$ between predictor and fitted values, where the mean expected response is given by Eq (1); (iv) It is possible for the variance to change with the covariates only as a function of mean given by Eq (2).

$$E(Y|X) = \mu = g^{-1}(X\beta) \tag{1}$$

$$Var(Y|X) = V(g^{-1}(X\beta)) \tag{2}$$

Density function of the Gaussian distribution, in terms of $\mu$ is given by $f(y; \mu, \sigma^2) = \frac{1}{\sqrt{2\pi\sigma^2}} exp\left\{-\frac{(y-\mu)^2}{2\sigma^2}\right\}$, where $f(\cdot)$ is the generic form of y density function. Please note that GLM optimize the deviance, which is $D = 2\phi\{\mathcal{L}(y, \sigma^2; y) - \mathcal{L}(\mu, \sigma^2; y)\}$, where the observed log-likelihood function is $\mathcal{L}(\mu, \sigma^2; y) = \sum_{i=1}^{n}\left\{\frac{y_i\mu_i - \mu_i^2/2}{\sigma^2} - \frac{y_i^2}{2\sigma^2} - \frac{1}{2}ln(2\pi\sigma^2)\right\}$ and saturated log-likelihood function is $\mathcal{L}(y, \sigma^2; y) = \sum_{i=1}^{n}\left\{\frac{y_i^2 - \mu_i^2/2}{\sigma^2} - \frac{y_i^2}{2\sigma^2} - \frac{1}{2}ln(2\pi\sigma^2)\right\}$. After simplifying deviance, $D = \sum_{i=1}^{n}(y_i - \mu_i)^2$, which is same as sum of squared residuals.

The temporal stability of the data is an essential requirement to achieve reliable statistical results. Therefore, the likelihood tests are conducted on the crash severity models to determine whether there is significant difference across the five studies time-periods. The log-likelihood test [36,37] is shown as;

$$\chi^2 = -2[LL(\beta_f) - LL(\beta_n)] \tag{3}$$

Where $LL(\beta_f)$ is the log-likelihood of the full or final model with all the independent variables and $LL(\beta_n)$ is the log-likelihood of restricted model for the same dependent variable. To determine the confidence level, based on which the null hypothesis (null model is better than the full model) is tested using $\chi^2$ test. For acceptance or rejection, a $p = 0.05$ is selected to reject null hypothesis. S2 Table shows the variables involved in this study as well as their code and remarks.

## Results

Table 1 presents a comprehensive overview of our dataset's descriptive statistics. Among the 802 road crashes recorded on the N-5, 481 were categorised as fatal crashes, while 321 were non-fatal. Notably, the data analysis reveals that a substantial 60% of these crashes were fatal,

**Table 1. Descriptive statistics of variables.**

| Variables | Number of crashes (n = 802) | Proportion of crashes % |
|---|---|---|
| **Crash year** | | |
| 2013 | 127 | 15.8 |
| 2014 | 130 | 16.2 |
| 2015 | 287 | 35.8 |
| 2016 | 152 | 18.9 |
| 2017 | 107 | 13.3 |
| **Crash day** | | |
| Day = 1 (Monday) | 107 | 13.3 |
| Day = 2 (Tuesday) | 99 | 12.3 |
| Day = 3 (Wednesday) | 111 | 13.9 |
| Day = 4 (Thursday) | 111 | 13.9 |
| Day = 5 (Friday) | 136 | 17.0 |
| Day = 6 (Saturday) | 117 | 14.6 |
| Day = 7 (Sunday) | 120 | 15.0 |
| **Cause of crash** | | |
| Causes of crash = 1 (Careless driving) | 220 | 27.4 |
| Causes of crash = 2 (Vehicle conditions) | 150 | 18.7 |
| Crash cause = 3 (Other factors) | 432 | 53.9 |
| **Vehicles involved** | | |
| Vehicle involved = 1 (3W) | 136 | 17 |
| Vehicle involved = 2 (LTV) | 305 | 38 |
| Vehicle involved = 3 (HTV) | 361 | 45 |
| **Season of the year** | | |
| Season = 1 (Hot) | 225 | 28.1 |
| Season 2 (Others) | 577 | 71.9 |
| **Time of the day** | | |
| Time = 1 (Daytime) | 494 | 61.6 |
| Time = 2 (Nighttime) | 308 | 38.4 |

**Table 2. Log likelihood ratio of different models.**

| Year | Log likelihood of actual model | Log likelihood of null model | $\chi^2$ | p-value |
|------|-------------------------------|------------------------------|----------|---------|
| 2013 | -25.98 | -42.306 | 41.44 | < .001 |
| 2014 | -12.393 | -112.08 | 199.374 | < .001 |
| 2015 | -41.315 | -236.631 | 390.632 | < .001 |
| 2016 | -25.98 | -176.722 | 301.484 | < .001 |
| 2017 | -22.222 | -58.211 | 71.978 | < .001 |

underscoring the severity of road safety concerns. It is essential to acknowledge that Pakistan faces challenges related to under-reporting of non-fatal crashes, particularly those resulting in minor injuries, and property-damage incidents that are often privately resolved are not included in our study. The preeminence of fatal crashes over non-fatal ones consistently persisted annually. Notably, the results indicate a decline in the number of crashes in 2013 and 2014, followed by a noteworthy surge in 2015. Moreover, data shows that Friday recorded the highest number of road crashes compared to other days of the week, and road crashes on weekends were significantly more frequent than on weekdays. Careless driving emerged as a leading cause of road crashes, accounting for 27.4% of incidents. Vehicle conditions also played a role, contributing to 18.7% of fatal crashes. In terms of the types of vehicles involved, Heavy Transport Vehicles (HTV) were slightly more associated with fatal road crashes (45%) in comparison to Light Transport Vehicles (LTV) at 38% and Three-Wheelers (3W) at 17%. The data also reveals that 28.1% of road crashes occurred during the hot weather, while a considerable majority transpired during daylight hours (61.6%) compared to night-time.

Table 2 presents the estimation results of models using 2013, 2014, 2015, 2016, and 2017 data, respectively. The results show that the parameters that are the same between two-time periods are rejected over 90% confidence. No test was found in which the test produces insignificant value. This shows that the model specifications and estimated parameters are significant temporal instable from year 2013 to 2017. All coefficients shown in the tables are statically significant at the 0.05 significance level.

Tables 3–7 presents the results of fitting the GLM accounting for different independent variables predicting the crash severity in years 2013–2017. Table 4 shows the modelling results of GLM for the year of 2013. It is revealed that days of the occurrence of crashes, beats, causes of crashes and vehicles involved in road crashes had a significant effect on crash severity (Model $\chi^2$ = 41.440, p<0.01). As compared to Sundays, the road crashes occurred on Thursday were found to be less severe in terms of death (β = -0.497**, OR = 0.61, p<0.01). Crashes occurred on beat 1, 2, 3 and 9 were found to be severe in terms of death. Keeping beat 9 as reference, crashes occurred on beat 1, 3 and 9 were found to be more severe than beat 9, whereas beat 2 was less severe than beat 9 in terms of death. The causes of road crashes were also found to be significant predictor of crash severity. Vehicle conditions were found to be less involved in crash severity than other causes of crash severity (β = -0.410**, OR = 0.66, p<0.01). This implies that the crashes caused by poor road conditions, poor visibility and wrong overtaking were found to be more severe in terms of death as compared to crashes caused by vehicle conditions. One of the significant factors which affects the crash severity is found to be vehicles involved in crashes. The results showed the crashes involving LTV are found to be more severe than 3W (β = -.564**, OR = 0.57, p<0.01) and HTV (β = -0.375**, OR = 0.69, p<0.01). Akaike's Information Criterion (AIC) and Bayesian Information Criterion (BIC) values are also mentioned in the subsequent tables.

In Table 4, the results of GLM of the 2014 model are given. Some of the results are in line with the results obtained from the 2013 model in which days, beat and causes of crashes were

**Table 3. Results of the GLM for identification of contributory variables to crash severity (2013).**

| Variable | | | B | SE | Wald | OR | 95% C.I | |
|---|---|---|---|---|---|---|---|---|
| | | | | | | | Lower bound | Upper bound |
| **2013** | | | | | | | | |
| Days | Day = 1 (Monday) | | .190 | .2295 | .684 | 1.21 | -.260 | .640 |
| | Day = 2 (Tuesday) | | .066 | .2173 | .092 | 1.07 | -.360 | .492 |
| | Day = 3 (Wednesday) | | -.039 | .2299 | .028 | 0.96 | -.489 | .412 |
| | Day = 4 (Thursday) | | **-.497**** | **.2486** | **3.991** | **0.61** | **-.984** | **-.009** |
| | Day = 5 (Friday) | | .042 | .1942 | .047 | 1.04 | -.338 | .423 |
| | Day = 6 (Saturday) | | -.094 | .2378 | .156 | 0.91 | -.560 | .372 |
| | Day = 7 (Sunday) | | 0ª | | | | | |
| Time | 1 (Daytime) | | -.052 | .2316 | .051 | 0.95 | -.506 | .402 |
| | 2 (Nighttime) | | 0ª | | | | | |
| Season | 1 (Hot) | | .198 | .1312 | 2.277 | 1.22 | -.059 | .455 |
| | 2 (Other) | | 0ª | | | | | |
| Beat | Beat 1 | | **.424*** | **.1789** | **5.612** | **1.53** | **.073** | **.774** |
| | Beat 2 | | **-.307*** | **.3625** | **.718** | **0.74** | **-1.018** | **.403** |
| | Beat 3 | | **.585*** | **.2887** | **4.101** | **1.79** | **.019** | **1.151** |
| | Beat 4 | | -.048 | .1616 | .087 | 0.95 | -.364 | .269 |
| | Beat 5 | | .279 | .2470 | 1.279 | 1.32 | -.205 | .763 |
| | Beat 6 | | .389 | .2624 | 2.199 | 1.48 | -.125 | .903 |
| | Beat 7 | | .160 | .2237 | .512 | 1.17 | -.278 | .598 |
| | Beat 8 | | -.196 | .4366 | .201 | 0.82 | -1.051 | .660 |
| | Beat 9 | | **.749*** | **.3699** | **4.103** | **2.11** | **.024** | **1.474** |
| | Beat 10 | | .200 | .2373 | .713 | 1.22 | -.265 | .666 |
| | Beat 11 | | 0ª | | | | | |
| Cause of crashes | 1 (Careless driving) | | .138 | .1655 | .690 | 1.15 | -.187 | .462 |
| | 2 (Vehicle conditions) | | **-.410*** | **.2485** | **2.725** | **0.66** | **-.897** | **.077** |
| | 3 (Other factors) | | 0ª | | | | | |
| Vehicles involved | 1 (3W) | | **-.564**** | **.1850** | **9.282** | **0.57** | **-.926** | **-.201** |
| | 2 (LTV) | | **-.375**** | **.1561** | **5.771** | **0.69** | **-.681** | **-.069** |
| | 3 (HTV) | | 0ª | | | | | |

Likelihood Ratio chi square = 41.440, df = 23, sig = .011.

Akaike's Information Criterion (AIC) = 93.172.

Bayesian Information Criterion (BIC) = 151.44.

B = Standard coefficients, OR = Odds Ratio, SE = Standard Error, C.I = Confidence Interval.

* >90% level of significance.

** >95% level of significance.

found to be significant predictors of crash severity. On the other hand, time of the occurrence of crash, weather and vehicles involved in the crashes were not found to be significant in 2014 model ($X^2$ = 71.509, p>.01). As compared to Sundays, the road crashes occurred on Wednesday were found to be less severe in terms of death (β = -0.391**, OR = 0.68, p<0.01). Crashes occurred on beat 1, 2, 3, 4 and 6 were found to be severe in terms of death. Crashes occurred on beat 1, 2, 3, 4 and 9 were found to be less severe than beat 11 in terms of death. In addition to that, the results of the model portray that crashes caused by poor road conditions, poor visibility and wrong overtaking were found to be less severe as compared to crashes caused by vehicle conditions.

**Table 4. Results of the GLM for identification of contributory variables to crash severity (2014).**

| Variable | | | B | SE | Wald | OR | 95% C.I | |
|---|---|---|---|---|---|---|---|---|
| | | | | | | | Lower bound | Upper bound |
| **2014** | | | | | | | | |
| Days | Day = 1 (Monday) | | -.311 | .1932 | 2.591 | 0.73 | -.690 | .068 |
| | Day = 2 (Tuesday) | | -.165 | .2197 | .564 | 0.85 | -.596 | .266 |
| | Day = 3 (Wednesday) | | **-.391*** | **.1817** | **4.635** | **0.68** | **-.747** | **-.035** |
| | Day = 4 (Thursday) | | -.072 | .1981 | .132 | 0.93 | -.460 | .316 |
| | Day = 5 (Friday) | | -.035 | .1843 | .036 | 0.97 | -.396 | .326 |
| | Day = 6 (Saturday) | | -.210 | .1736 | 1.456 | 0.81 | -.550 | .131 |
| | Day = 7 (Sunday) | | 0[a] | | | | | |
| Time | 1 (Daytime) | | -.176 | .1964 | .800 | 0.84 | -.560 | .209 |
| | 2 (Nighttime) | | 0[a] | | | | | |
| Weather | 1 (Hot) | | **-.231**** | **.1131** | **4.177** | **0.79** | **-.453** | **-.009** |
| | 2 (Other) | | 0[a] | | | | | |
| Beat | Beat 1 | | **-2.280**** | **1.0788** | **4.465** | **0.10** | **-4.394** | **-.165** |
| | Beat 2 | | **-2.204**** | **.9741** | **5.117** | **0.11** | **-4.113** | **-.294** |
| | Beat 3 | | **-1.659*** | **.8879** | **3.489** | **0.19** | **-3.399** | **.082** |
| | Beat 4 | | **-1.555**** | **.7833** | **3.939** | **0.21** | **-3.090** | **-.019** |
| | Beat 5 | | -.917 | .6206 | 2.185 | 0.40 | -2.133 | .299 |
| | Beat 6 | | **-1.125*** | **.5632** | **3.992** | **0.32** | **-2.229** | **-.021** |
| | Beat 7 | | -.472 | .4491 | 1.104 | 0.62 | -1.352 | .408 |
| | Beat 8 | | .334 | .3838 | .757 | 1.40 | -.418 | 1.086 |
| | Beat 9 | | -.061 | .2911 | .044 | 0.94 | -.632 | .509 |
| | Beat 10 | | .048 | .1957 | .061 | 1.05 | -.335 | .432 |
| | Beat 11 | | 0[a] | | | | | |
| Cause of crashes | Causes of crash = 1 (Careless driving) | | -.001 | .1189 | .000 | 1.00 | -.235 | .232 |
| | Causes of crash = 2 (Vehicle conditions) | | **.859**** | **.1987** | **18.673** | **2.36** | **.469** | **1.248** |
| | Crash cause = 3 (Other factors) | | 0[a] | | | | | |
| Vehicles involved | 1 (3W) | | -.223 | .1497 | 2.226 | 0.80 | -.517 | .070 |
| | 2 (LTV) | | -.009 | .1292 | .005 | 0.99 | -.262 | .244 |
| | 3 (HTV) | | 0[a] | | | | | |

Likelihood Ratio chi square = 71.509, df = 23, sig < .001.

Akaike's Information Criterion (AIC) = 74.786.

Bayesian Information Criterion (BIC) = 133.703.

B = Standard coefficients, OR = Odds Ratio, SE = Standard Error, C.I = Confidence Interval.

\* >90% level of significance.

\*\*>95% level of significance.

Table 5 presents the results of the GLM employed to explore the factors affecting crash severity in year 2015 ($\chi^2$ = 53.279, p<0.01). The results revealed that crashes on Fridays are more severe as compared to Sundays ($\beta$ = 0.282**, OR = 1.33, p<0.01). As compared to other weathers, the crashes in hot weather were found to be more severe in 2015 ($\beta$ = 0.203**, OR = 1.23, p<0.01). Crashes caused by aggressive driving, dozing at wheels and distracted driving were found to be less severe in terms of severity as compared to crashes caused by poor road conditions, poor visibility and wrong overtaking ($\beta$ = 0.182**, OR = 0.83, p < .001). In addition to that, crashes in 3W ($\beta$ = -0.404**, OR = 0.67, p<0.01) and HTVs ($\beta$ = -0.204**, OR = 0.82, p<0.01) were found to less severe as compared to the LTVs.

**Table 5. Results of the GLM for identification of contributory variables to crash severity (2015).**

| Variable | | | B | SE | Wald | OR | 95% C.I | |
|---|---|---|---|---|---|---|---|---|
| | | | | | | | Lower bound | Upper bound |
| **2015** | | | | | | | | |
| Days | Day = 1 (Monday) | | .158 | .1218 | 1.673 | 1.17 | -.081 | .396 |
| | Day = 2 (Tuesday) | | -.138 | .1284 | 1.163 | 0.87 | -.390 | .113 |
| | Day = 3 (Wednesday) | | -.066 | .1240 | .282 | 0.94 | -.309 | .177 |
| | Day = 4 (Thursday) | | .197 | .1263 | 2.436 | 1.22 | -.050 | .445 |
| | Day = 5 (Friday) | | **.282**\*\* | **.1355** | **4.340** | **1.33** | **.017** | **.548** |
| | Day = 6 (Saturday) | | -.041 | .1257 | .106 | 0.96 | -.287 | .206 |
| | Day = 7 (Sunday) | | 0[a] | | | | | |
| Time | 1 (Daytime) | | .070 | .0744 | .893 | 1.07 | -.075 | .216 |
| | 2 (Nighttime) | | 0a | | | | | |
| Season | 1 (Hot) | | **.203**\*\* | **.0893** | **5.195** | **1.23** | **.028** | **.378** |
| | 2 (Other) | | 0a | | | | | |
| Beat | Beat 1 | | -.100 | .1921 | .270 | 0.90 | -.476 | .277 |
| | Beat 2 | | -.312 | .2214 | 1.988 | 0.73 | -.746 | .122 |
| | Beat 3 | | -.004 | .1735 | .001 | 1.00 | -.344 | .336 |
| | Beat 4 | | .002 | .1388 | .000 | 1.00 | -.270 | .274 |
| | Beat 5 | | .041 | .1339 | .093 | 1.04 | -.222 | .303 |
| | Beat 6 | | -.018 | .1417 | .017 | 0.98 | -.296 | .259 |
| | Beat 7 | | .057 | .1421 | .161 | 1.06 | -.222 | .336 |
| | Beat 8 | | .163 | .1403 | 1.352 | 1.18 | -.112 | .438 |
| | Beat 9 | | -.100 | .2112 | .225 | 0.90 | -.514 | .314 |
| | Beat 10 | | .134 | .1895 | .504 | 1.14 | -.237 | .506 |
| | Beat 11 | | 0a | | | | | |
| Cause of crashes | 1 (Careless driving) | | **-.182**\*\* | **.0844** | **4.642** | **0.83** | **-.347** | **-.016** |
| | 2 (Vehicle conditions) | | -.180 | .1571 | 1.309 | 0.84 | -.488 | .128 |
| | 3 (Other factors) | | 0[a] | | | | | |
| Vehicles involved | 1 (3W) | | **-.404**\*\* | **.0892** | **20.541** | **0.67** | **-.579** | **-.229** |
| | 2 (LTV) | | **-.204**\*\* | **.0839** | **5.927** | **0.82** | **-.369** | **-.040** |
| | 3 (HTV) | | 0[a] | | | | | |

Likelihood Ratio chi square = 53.279, df = 24, sig < .001.

Akaike's Information Criterion (AIC) = 134.631.

Bayesian Information Criterion (BIC) = 216.314.

B = Standard coefficients, OR = Odds Ratio, SE = Standard Error, C.I = Confidence Interval.

\* >90% level of significance.

\*\*>95% level of significance.

In the 2016 model (Table 6), only season was found to be the significant factor associated with crash severity ($\chi^2$ = 53.279, p>0.01). The results reveal that as compared to other seasons, hot weather was found to be less severe in terms of fatalities ($\beta$ = -0.220\*\*, OR = 0.80, p<0.01). Surprisingly, no other factors such as days, time, beat, causes and vehicles involved in road crashes were found to be significant predictors of crash severity.

In Table 7, the modelling results of GLM for the year of 2017 are presented. In 2017 model, days, causes of crashes and vehicles involved in road crashes were found significant at a 95% confidence interval ($\chi^2$ = 57.707, p<0.01). The road crashes occurred on Mondays were found to 0.59 times more fatal than crashes occurred on Sundays. Furthermore, Thursday and Saturday were found to be 0.61 and 0.52 times fatal as on Sundays. In addition to that, crashes

**Table 6. Results of the GLM for identification of contributory variables to crash severity (2016).**

| Variable | | | B | SE | Wald | OR | 95% C.I | |
|---|---|---|---|---|---|---|---|---|
| | | | | | | | Lower bound | Upper bound |
| **2016** | | | | | | | | |
| Days | Day = 1 (Monday) | | .094 | .2255 | .174 | 1.10 | -.348 | .536 |
| | Day = 2 (Tuesday) | | -.402 | .1887 | 4.540 | 0.67 | -.772 | -.032 |
| | Day = 3 (Wednesday) | | -.511 | .1841 | 7.708 | 0.60 | -.872 | -.150 |
| | Day = 4 (Thursday) | | -.481 | .1865 | 6.638 | 0.62 | -.846 | -.115 |
| | Day = 5 (Friday) | | -.294 | .1847 | 2.539 | 0.75 | -.656 | .068 |
| | Day = 6 (Saturday) | | -.437 | .2136 | 4.177 | 0.65 | -.855 | -.018 |
| | Day = 7 (Sunday) | | 0[a] | | | | | |
| Time | 1 (Daytime) | | -.038 | .1362 | .076 | 0.96 | -.304 | .229 |
| | 2 (Nighttime) | | 0[a] | | | | | |
| Season | 1 (Hot) | | **-.220*** | **.1202** | **3.352** | **0.80** | **-.456** | **.016** |
| | 2 (Other) | | 0[a] | | | | | |
| Beat | Beat 1 | | .306 | .8149 | .141 | 1.36 | -1.291 | 1.903 |
| | Beat 2 | | .097 | .7178 | .018 | 1.10 | -1.310 | 1.503 |
| | Beat 3 | | -.106 | .6280 | .029 | 0.90 | -1.337 | 1.124 |
| | Beat 4 | | -.164 | .5279 | .096 | 0.85 | -1.198 | .871 |
| | Beat 5 | | .410 | .4876 | .708 | 1.51 | -.545 | 1.366 |
| | Beat 6 | | .147 | .3649 | .163 | 1.16 | -.568 | .863 |
| | Beat 7 | | -.278 | .5293 | .276 | 0.76 | -1.315 | .759 |
| | Beat 8 | | -.007 | .2616 | .001 | 0.99 | -.519 | .506 |
| | Beat 9 | | .017 | .2066 | .007 | 1.02 | -.388 | .422 |
| | Beat 10 | | .306 | .8149 | .141 | 1.36 | -1.291 | 1.903 |
| | Beat 11 | | 0[a] | | | | | |
| Cause of crashes | 1 (Careless driving) | | -.198 | .1431 | 1.912 | 0.82 | -.478 | .083 |
| | 2 (Vehicle conditions) | | .272 | .2317 | 1.373 | 1.31 | -.183 | .726 |
| | 3 (Other factors) | | 0[a] | | | | | |
| Vehicles involved | 1 (3W) | | -.108 | .1498 | .520 | 0.90 | -.402 | .186 |
| | 2 (LTV) | | -.111 | .1417 | .609 | 0.89 | -.388 | .167 |
| | 3 (HTV) | | 0[a] | | | | | |

Likelihood Ratio chi square = 60.319, df = 23, sig < .001.

Akaike's Information Criterion (AIC) = 101.96.

Bayesian Information Criterion (BIC) = 164.732.

B = Standard coefficients, OR = Odds Ratio, SE = Standard Error, C.I = Confidence Interval.

* >90% level of significance.

** >95% level of significance.

caused by aggressive driving, dozing at wheels and distracted driving were found to be less severe in terms of death as compared to crashes caused by poor road conditions, poor visibility, and wrong overtaking (β = -.369**, OR = 0.69, p<0.01). Furthermore, the crashes in 3W and HTV were found to be 0.50 and 0.53 times fatal than the crashes in LTV. The values of other variables could not achieve significant values.

Table 8 provides the list of factors having temporal stability of crash severity in Pakistan. The results clearly show that with the passage of time, several contributors were found to be insignificant. Overall, the results concluded that there is a temporal instability of factors contributing to crash severity in Pakistan.

**Table 7. Results of the GLM for identification of contributory variables to crash severity (2017).**

| Variable | | B | SE | Wald | OR | 95% C.I | |
|---|---|---|---|---|---|---|---|
| | | | | | | Lower bound | Upper bound |
| **2017** | | | | | | | |
| Days | Day = 1 (Monday) | **-.522**** | **.2210** | **5.577** | **0.59** | **-.955** | **-.089** |
| | Day = 2 (Tuesday) | -.460 | .3653 | 1.585 | 0.63 | -1.176 | .256 |
| | Day = 3 (Wednesday) | -.282 | .3408 | .687 | 0.75 | -.950 | .385 |
| | Day = 4 (Thursday) | **-.496**** | **.2227** | **4.955** | **0.61** | **-.932** | **-.059** |
| | Day = 5 (Friday) | .075 | .2286 | .109 | 1.08 | -.373 | .523 |
| | Day = 6 (Saturday) | **-.657**** | **.2261** | **8.441** | **0.52** | **-1.100** | **-.214** |
| | Day = 7 (Sunday) | 0[a] | | | | | |
| Time | 1 (Daytime) | -.240 | .1682 | 2.033 | 0.79 | -.570 | .090 |
| | 2 (Nighttime) | 0[a] | | | | | |
| Season | 1 (Hot) | .061 | .1903 | .103 | 1.06 | -.312 | .434 |
| | 2 (Other) | 0[a] | | | | | |
| Beat | Beat 1 | -1.824 | 2.7851 | .429 | 0.16 | -7.283 | 3.635 |
| | Beat 2 | -1.460 | 2.5412 | .330 | 0.23 | -6.441 | 3.521 |
| | Beat 3 | -.581 | 2.1662 | .072 | 0.56 | -4.827 | 3.665 |
| | Beat 4 | -.889 | 1.9360 | .211 | 0.41 | -4.683 | 2.905 |
| | Beat 5 | -.975 | 1.5618 | .389 | 0.38 | -4.036 | 2.086 |
| | Beat 6 | .021 | 1.2485 | .000 | 1.02 | -2.426 | 2.468 |
| | Beat 7 | -.271 | .7910 | .117 | 0.76 | -1.821 | 1.279 |
| | Beat 8 | -1.110 | .7380 | 2.264 | 0.33 | -2.557 | .336 |
| | Beat 9 | .358 | .3386 | 1.116 | 1.43 | -.306 | 1.021 |
| | Beat 10 | -1.824 | 2.7851 | .429 | 0.16 | -7.283 | 3.635 |
| | Beat 11 | 0[a] | | | | | |
| Cause of crashes | 1 (Careless driving) | **-.369*** | **.1732** | **4.531** | **0.69** | **-.708** | **-.029** |
| | 2 (Vehicle conditions) | -.296 | .2403 | 1.516 | 0.74 | -.767 | .175 |
| | 3 (Other factors) | 0[a] | | | | | |
| Vehicles involved | 1 (3W) | **-.691**** | **.1970** | **12.313** | **0.50** | **-1.077** | **-.305** |
| | 2 (LTV) | **-.627**** | **.1927** | **10.582** | **0.53** | **-1.005** | **-.249** |
| | 3 (HTV) | 0[a] | | | | | |

Likelihood Ratio chi square = 57.707.509, df = 16, sig < .001.

Akaike's Information Criterion (AIC) = 94.444.

Bayesian Information Criterion (BIC) = 148.416.

B = Standard coefficients, OR = Odds Ratio, SE = Standard Error, C.I = Confidence Interval.

* >90% level of significance.

** >95% level of significance.

**Table 8. Significant factors associated with crash severity.**

| | Day of the week | Time of the day | Season | Beat number of road | Cause of the crash | Nature of the vehicle |
|---|---|---|---|---|---|---|
| 2013 | ✓ | ✗ | ✗ | ✓ | ✓ | ✓ |
| 2014 | ✓ | ✗ | ✓ | ✓ | ✓ | ✗ |
| 2015 | ✓ | ✗ | ✓ | ✗ | ✓ | ✓ |
| 2016 | ✗ | ✗ | ✓ | ✗ | ✗ | ✗ |
| 2017 | ✓ | ✗ | ✗ | ✗ | ✗ | ✓ |

## Discussions and conclusions

This paper aims at investigating the temporal stability of factors contributing to road crashes in Pakistan over the five years period. The motivation behind this study came from the high crash severity in Pakistan due to the mixed nature of traffic such as cars, jeeps, buses, trucks, motorcycles, rickshaws, and heavy vehicles. This study attempted to find various key risk factors contributing to crash severity in Pakistani drivers. This study provides two noteworthy contributions to the literature which were never investigated before. Firstly, it identifies the factors contributing to crash severity in Pakistani drivers and provides a vivid picture of road safety issues within the country. Secondly, it investigates the temporal stability of factors contributing to crash severity. GLM is employed to explore the significant factors affecting the crash severity for the five individual models. To determine if the developed GLM models are temporal stable across the investigated time, a set of likelihood ratio tests are also carried out. The results showed that the model specifications and estimated parameters were significant temporal instable from year 2013 to 2017. The day of the week, the location of the crashes, weather conditions, causes of the crashes, and the types of vehicles involved in road crashes were found to be significant factors contributing crash severity having temporal instability.

With respect to the factors contributing to crash severity in Pakistani drivers, this study reveals that days of the week involved in road crashes are the significant factors contributing crash severity with temporal instability. This study is somehow in consistent with a previous study in which authors reported that crash severity gradually increased from Tuesday to Sunday [38]. It is also found that road crashes on weekend tended to be more severe [39]. The temporal stability of crash patterns associated with days of the week holds critical implications for road safety planning and policy. These temporal trends may stem from diverse factors, encompassing driver behaviour, traffic patterns, and road infrastructure. For instance, on specific days, like weekends or holidays, individuals may be prone to engaging in risky behaviours like speeding or driving under the influence, resulting in consistent crash patterns. Likewise, heightened traffic congestion on certain days can amplify the risk of crashes, given reduced visibility and slower reaction times. These insights offer valuable input for contemporary road safety measures.

Weather is found to be the significant factor contributing to crash severity. Crash severity was found to be highest in hot weather in 2015 but due to temporal instability, it changed to low as more to other weather types. The possible explanation is that the summer in Pakistan is becoming severe every year due to climatic change which resulted in a significant reduction in traffic flow during peak summer in Pakistan. Generally, the reduction in traffic flow encourages the driver to speed faster, which may lead to severe road crashes. This finding is supported by a previous study where high speed of vehicle increases the crash severity [40]. The moderate temperature encourages commuters to travel outside their home city to enjoy pleasant weather. Furthermore, an abrupt elevation in the number of domestic and international tourist in moderate temperature increase the traffic flow and ultimately, increase the chances of road crashes. This finding is in line with a previous study [41], but in contrast with a study conducted in India [42]. Temporal stability of road crashes may be influenced by broader social, economic, or environmental factors that remain relatively constant over time. For example, certain regions or populations may be more susceptible to weather-related road crashes due to factors such as higher rates of vehicle ownership, less access to public transportation, or less experience driving in adverse weather conditions.

Careless driving and vehicle conditions were found to be significant factors contributing to crash severity as well as showed temporal instability. In careless driving, aggressive driving, untrained driving, and drowsy driving are the main causes included in this study. This study is

in line with a previous study in which aggressive driving was found to be significantly related to fatal crashes [43,44]. Regarding untrained driving, this finding aligns with a previous study in which driving without proper driving training was found to be significant predictor of road crashes in Pakistan [6]. This factor could be rectifying to normal by providing the driver with proper driving education and knowledge. Drowsy driving is associated with sleepiness and dozing at the wheel. The finding from this paper supports existing studies that long hour driving contributes to fatigue and sleepiness, leading to road crashes [45].

Furthermore, the findings of this study regarding the lack of temporal nature of vehicles over period of five years reveals that LTV are responsible for more severe as compared to 3W and HTV, despite having some temporal instability. This study aligns with one of our previous studies in which road crashes caused by careless driving, speeding, and poor road conditions were found to be more prevalent in small cars and motorcycles as compared to heavy trucks [46]. The possible explanation is that there are significant changes in the types of vehicles on the road over time, such as an increase in the number of large trucks, more fuel-efficient cars, which potentially impacted the frequency and severity of crashes, and may result in less temporal stability. Overall, the relationship between the nature of vehicles and temporal stability of road crashes is complex and may be influenced by a variety of factors related to vehicle safety features, maintenance, and changes in the types of vehicles on the road over time.

In addition to presenting the study findings, it's important to acknowledge the study's limitations. Firstly, the dataset used focuses only on one highway, N-5, leaving out information from other interconnected roads and highways across Pakistan. This limits the dataset's scope to cover all crashes comprehensively. Additionally, the dataset lacks traffic volume details, crucial for a highway like N-5, connecting various cities with diverse traffic patterns. This variety makes it challenging to precisely determine traffic volume. Although statistical methods were used for data analysis, potential bias in interpreting the results should be recognised. Furthermore, when modeling the effects of different variables on crash, different coefficients are estimated for the same variable across different years. To overcome the model uncertainty, it is recommended to consider the Bayesian model averaging approach [47] in future studies, which consider the effect of different variables.

The results of this study could aid decision-makers in identifying the factors that consistently contribute to an increased risk of crash severity of road crashes, allowing for targeted allocation of resources to mitigate the effects of these factors. The results of this research offer specific quantitative measures that can aid in assessing the practicality of implementing various strategies aimed at reducing the risk of crash severity of road crashes in Pakistani drivers. Considering the existing findings, some immediate road safety countermeasures are hereby suggested to minimise the life losses in road safety. It is necessary to design and conduct road safety awareness campaigns to raise awareness about the importance of careless driving. There is an urgent need to implement strict rules and regulations relevant to speed limits in day and night-time driving. It is already suggested in a previous study that improving the road infrastructure to separate the motor vehicle physically and road users might be helpful to mitigate road crashes [48]. Advanced Driver Assistance Systems (ADAS) can contribute to a reduction in traffic risk. ADAS typically includes technologies and features designed to enhance vehicle safety and assist drivers in various driving situations. These systems may offer functionalities such as collision avoidance, lane-keeping assistance, adaptive cruise control, and other features that can collectively contribute to a safer driving environment, ultimately lowering the risk of traffic crashes [49,50]. The wide spread introduction of vehicles with ADAS technologies is also a viable solution to reduce crash severity. In countries with mixed traffic like Pakistan, increasing the lateral gap between larger vehicles like trucks and trailers and smaller vehicles like cars and jeeps is one of the significant countermeasures to reduce road crashes. We believe

that this study has shed light on the importance the road safety to achieve the goal of safe transportation system in Pakistan.

## Supporting information

**S1 Table. Categorisation of contributory factors (causes of crash) involved in this study.**
(DOCX)

**S2 Table. Independent variable code and remarks.**
(DOCX)

## Acknowledgments

We acknowledge the National Highway and Motorway Police, Pakistan for the provision of road crash data.

## Author Contributions

**Conceptualization:** Abdulaziz H. Alshehri, Muhammad Hussain, Danish Farooq, Etikaf Hussain.

**Data curation:** Abdulaziz H. Alshehri, Muhammad Hussain, Danish Farooq.

**Formal analysis:** Abdulaziz H. Alshehri, Amjad Pervez, Muhammad Hussain.

**Funding acquisition:** Abdulaziz H. Alshehri, Muhammad Hussain.

**Investigation:** Muhammad Hussain, Danish Farooq, Etikaf Hussain.

**Methodology:** Abdulaziz H. Alshehri, Amjad Pervez, Etikaf Hussain.

**Project administration:** Amjad Pervez.

**Supervision:** Muhammad Hussain.

**Visualization:** Etikaf Hussain.

**Writing – original draft:** Abdulaziz H. Alshehri, Amjad Pervez, Muhammad Hussain, Danish Farooq, Etikaf Hussain.

**Writing – review & editing:** Abdulaziz H. Alshehri, Muhammad Hussain, Danish Farooq, Etikaf Hussain.

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
