## [Decision Letter · Decision Letter 0]

28 Dec 2023

PONE-D-23-42042Examination of factors associated with the temporal stability assessment of crash severity by using Generalised Linear Model-A case studyPLOS ONE

Dear Dr. HUSSAIN,

Thank you for submitting your manuscript to PLOS ONE. After careful consideration, we feel that it has merit but does not fully meet PLOS ONE’s publication criteria as it currently stands. Therefore, we invite you to submit a revised version of the manuscript that addresses the points raised during the review process.

We look forward to receiving your revised manuscript.

Kind regards,

Yajie Zou

Academic Editor

PLOS ONE

Journal Requirements:

   "No competing interest"

5. Please provide a complete Data Availability Statement in the submission form, ensuring you include all necessary access information or a reason for why you are unable to make your data freely accessible. If your research concerns only data provided within your submission, please write "All data are in the manuscript and/or supporting information files" as your Data Availability Statement.

6. We note that Figure 3 in your submission contain map/satellite images which may be copyrighted. All PLOS content is published under the Creative Commons Attribution License (CC BY 4.0), which means that the manuscript, images, and Supporting Information files will be freely available online, and any third party is permitted to access, download, copy, distribute, and use these materials in any way, even commercially, with proper attribution. For these reasons, we cannot publish previously copyrighted maps or satellite images created using proprietary data, such as Google software (Google Maps, Street View, and Earth). For more information, see our copyright guidelines: http://journals.plos.org/plosone/s/licenses-and-copyright.

a. You may seek permission from the original copyright holder of Figure 3 to publish the content specifically under the CC BY 4.0 license.  

Reviewers' comments:

Reviewer's Responses to Questions

**Comments to the Author**

1. Is the manuscript technically sound, and do the data support the conclusions?

Reviewer #1: Partly

Reviewer #2: Yes

2. Has the statistical analysis been performed appropriately and rigorously? 

Reviewer #1: Yes

Reviewer #2: Yes

3. Have the authors made all data underlying the findings in their manuscript fully available?

Reviewer #1: Yes

Reviewer #2: Yes

4. Is the manuscript presented in an intelligible fashion and written in standard English?

Reviewer #1: Yes

Reviewer #2: Yes

5. Review Comments to the Author

Reviewer #1: The authors are suggested to carefully address the comments below:

- The paper writing should be significantly improved.

- In the literature review section, many recent methods for analyzing crash severity are missing.

- When modeling the effect of different variables on crash, different coefficients are estimated for the same variable across different years. To overcome the model uncertainty, the authors are suggested review the Bayesian model averaging approach which can consider the effect of different variables. For example, see: Application of Bayesian model averaging for modeling time headway distribution. Physica A: Statistical Mechanics and Its Applications, 620, 128747. Improving transferability of safety performance functions by Bayesian model averaging. Transportation research record, 2280(1), 162-172. This point should be discussed as the future work.

- Why do the authors use the GLM model in this study? How about other modeling approaches?

- It seems that Year 2013 and Year 2017 do not have sufficient data observations for the analysis in this study.

- Why not aggregate all observations into one single dataset and then apply the analysis approach?

Reviewer #2: 1. The total sample size should be explained in Table 1.

2. Why do the authors build models for different years? Does single year data contain enough samples for regression analysis?

3. For the regression results, some variables are insignificant.

4. For Table 8, how to explain the difference in the modeling results?

5. What are the practical contributions of this study?

6. The traffic risk can be reduced with the development of ADAS, this point should also be discussed in the study. And some recent studies about ADAS and traffic risk should be reviewed. For example, see: Spatiotemporal Interaction Pattern Recognition and Risk Evolution Analysis During Lane Changes. IEEE Transactions on Intelligent Transportation Systems. Traffic Risk Assessment Based on Warning Data. Journal of Advanced Transportation, 2022.

6. PLOS authors have the option to publish the peer review history of their article (what does this mean?). If published, this will include your full peer review and any attached files.

Reviewer #1: No

Reviewer #2: No

---

## [Decision Letter · Decision Letter 1]

6 Feb 2024

Examination of factors associated with the temporal stability assessment of crash severity by using Generalised Linear Model-A case study

PONE-D-23-42042R1

Dear Dr. HUSSAIN,

We’re pleased to inform you that your manuscript has been judged scientifically suitable for publication and will be formally accepted for publication once it meets all outstanding technical requirements.

Kind regards,

Yajie Zou

Academic Editor

PLOS ONE

Additional Editor Comments (optional):

Reviewers' comments:

Reviewer's Responses to Questions

**Comments to the Author**

1. If the authors have adequately addressed your comments raised in a previous round of review and you feel that this manuscript is now acceptable for publication, you may indicate that here to bypass the “Comments to the Author” section, enter your conflict of interest statement in the “Confidential to Editor” section, and submit your "Accept" recommendation.

Reviewer #1: (No Response)

Reviewer #2: All comments have been addressed

2. Is the manuscript technically sound, and do the data support the conclusions?

Reviewer #1: (No Response)

Reviewer #2: Yes

3. Has the statistical analysis been performed appropriately and rigorously? 

Reviewer #1: (No Response)

Reviewer #2: Yes

4. Have the authors made all data underlying the findings in their manuscript fully available?

Reviewer #1: (No Response)

Reviewer #2: Yes

5. Is the manuscript presented in an intelligible fashion and written in standard English?

Reviewer #1: (No Response)

Reviewer #2: Yes

6. Review Comments to the Author

Reviewer #1: (No Response)

Reviewer #2: (No Response)

7. PLOS authors have the option to publish the peer review history of their article (what does this mean?). If published, this will include your full peer review and any attached files.

Reviewer #1: No

Reviewer #2: No

---

## [Editor Report · Acceptance letter]

8 Apr 2024

PONE-D-23-42042R1 

PLOS ONE

Dear Dr. Hussain, 

I'm pleased to inform you that your manuscript has been deemed suitable for publication in PLOS ONE. Congratulations! Your manuscript is now being handed over to our production team.

Kind regards, 

on behalf of

Dr. Yajie Zou 

Academic Editor

PLOS ONE